# Minimally Invasive Salvage Approaches for Management of Recurrence After Primary Renal Mass Ablation

**DOI:** 10.3390/cancers17060974

**Published:** 2025-03-13

**Authors:** Mohammadreza Askarpour, Alireza Aminsharifi

**Affiliations:** 1Department of Urology, University of Pittsburgh School of Medicine, Pittsburgh, PA 15232, USA; askarvip2@gmail.com; 2Department of Urology, Hershey Medical Center, PennState College of Medicine, Hershey, PA 17033, USA

**Keywords:** renal cell carcinoma, small renal mass, recurrence, ablation, salvage nephrectomy, active surveillance, minimally invasive surgery, laparoscopic, robotic

## Abstract

Ablative modalities such as cryoablation (CA), microwave ablation (MWA), and radiofrequency ablation (RFA) offer effective treatment for small renal masses in carefully selected patients. However, tumor recurrence after primary ablation remains a significant challenge. Currently, there is no consensus on the optimal minimally invasive salvage approach for managing recurrence after primary ablation. This review includes the largest accumulated series of patients to date with recurrence of cancer after primary renal ablation. We present the outcomes of various minimally invasive strategies, including repeat ablation, laparoscopic or robotic partial/radical nephrectomy, and active surveillance. By providing a comprehensive understanding of these approaches, this review supports risk-stratified clinical decision-making to enhance outcomes for patients experiencing tumor recurrence after renal mass ablation.

## 1. Introduction

Kidney cancer, now the 12th most common malignancy worldwide, is on the rise, with an estimated 400,000 new cases and 175,000 deaths annually. Around 50% of new patients are diagnosed with stage I renal cell carcinoma (RCC) (T1a), with reports showing an increase in the prevalence of this rate from 43% to 57% between 1993 and 2004 [1]. A small renal mass (SRM) is consistent with the definition of clinical stage I kidney tumors (cT1a) as incidentally detected, small-size (<4 cm), contrast-enhancing solid or cystic lesions [2]. When approaching patients with SRM, nephron-sparing procedures should be prioritized. Ablative modalities have emerged as a viable alternative for managing cT1a solid renal masses, particularly those smaller than 3 cm, as recommended by the American Urological Association (AUA) guidelines [3]. These therapies demonstrate favorable efficacy and excellent safety profiles, contributing to their growing utilization. With advantages such as low postoperative morbidity and renal function preservation, ablative techniques are confidently employed by radiologists and urologists through percutaneous and laparoscopic approaches [4,5]. Common ablation modalities include cryoablation (CA), microwave ablation (MWA), radiofrequency ablation (RFA), high-intensity focused ultrasound (HIFU), and irreversible electroporation (IRE) [6,7].

Recurrence rates following ablative techniques vary, ranging from 1 to 9% based on the method and approach employed [8]. CA and RFA show comparable recurrence rates, with CA exhibiting local recurrence rates of about 7.2%. RFA recurrence rates range from 4.2% in T1a to 14.3% in T1b tumors, with tumor stage being a key predictor of recurrence risk. Microwave ablation (MWA) demonstrates lower recurrence rates (2–5% at 1 year) compared to CA but similar rates at 5 years, likely due to its higher intertumoral temperature and larger ablation zone [8]. Most recurrences are detected within the first five years after the procedure [9]. Given the higher recurrence rates associated with ablative therapies compared to partial or radical nephrectomy, it is crucial to establish a well-structured follow-up strategy and provide salvage modalities to address the treatment failures [3].

There is currently no consensus on the optimal treatment for management of tumor recurrence following ablative therapies. Re-ablation, partial or radical nephrectomy, and active surveillance remain the primary management strategies employed to address post-ablation tumor recurrence [10]. Re-ablation is commonly used due to its demonstrated efficacy and acceptable complication rates, particularly in older patients with higher comorbidities [11]. Despite technical difficulties posed by fibrosis from primary ablation treatments, laparoscopic/robotic partial and radical nephrectomy remain viable treatment options in selected cases [12].

This literature review aims to present the outcomes of various minimally invasive approaches for the management of tumor recurrence following the primary ablation of small renal masses. To our knowledge, this is the largest accumulated series of patients with recurrence after renal mass ablation who were managed with minimally invasive modalities (including active surveillance).

## 2. Material and Methods

A systematic literature review was conducted using the Medline (PubMed) database, adhering to the 2020 Preferred Reporting Items for Systematic Reviews and Meta-analyses (PRISMA) protocol for reporting [13]. The protocol has not been registered. Various combinations of the following keywords were used to identify relevant articles: salvage, therapy, recurrent, kidney, cancer, ablation, renal retreatment, mass, ablative, nephrectomy, MWA, RFA, HIFU, CA, partial, radical, robotic, percutaneous, laparoscopic, and minimally invasive surgery. The search results were filtered by species (human), publication type (article), language (English), and publication year (1981 to 2024).

Studies were selected based on the PICO criteria: [patient, intervention, comparator, outcome] [14].

(P) Adults (>18 years) with prior ablative therapy for renal cancer presenting with tumor recurrence during follow-up period.(I) Minimally invasive treatment for tumor recurrence after primary ablation, including re-ablation, laparoscopic or robotic partial or radical nephrectomy, and active surveillance.(C) Comparisons between different re-treatment modalities regarding outcomes, follow-up, and other relevant factors.(O) Outcomes of interest included primary and secondary treatment success, retreatment outcomes (e.g., disease recurrence, metastasis, death, successful treatment or no evidence of disease (NED)), tumor types and stages, time to recurrence, follow-up duration, and patient characteristics like age and gender, if available.

Figure 1 illustrates the study selection process following the PRISMA protocol. The initial search identified 2064 studies published between 1981 and 2024 using various keyword combinations. Sixty additional studies were included after reviewing the reference lists of relevant studies to identify further management strategies for recurrence after ablative therapies and their outcomes. Of these, 14 were excluded due to duplication. After screening titles and abstracts for our eligibility criteria, 1945 articles were excluded. The full texts of the remaining 165 articles were studied; an additional 136 studies were excluded. These exclusions encompassed studies with an open surgery approach for salvage treatments, review articles, editorial comments, editor letters, and conference abstracts. Moreover, 35 additional studies were excluded despite meeting our initial criteria, due to missed critical data, such as the procedure or patients’ outcomes [15,16,17,18,19,20,21,22,23,24,25,26,27,28,29,30,31,32,33,34,35,36,37,38,39,40,41,42,43,44,45,46,47,48,49]. A full-text review of the remaining studies resulted in the inclusion of 29 studies, which collectively evaluated 364 patients [12,50,51,52,53,54,55,56,57,58,59,60,61,62,63,64,65,66,67,68,69,70,71,72,73,74,75,76,77]. Out of the 29 included studies, 24 were retrospective cohort studies, 4 were prospective cohort studies, and 1 was a case series.

Demographic data, including the type of initial ablative therapy; the time interval between the first treatment and recurrence; patient age; the type of secondary retreatment management, tumor pathology, and stage; re-treatment outcomes; and follow-up duration, were systematically collected. These data formed the basis of our analysis and enabled a comparative evaluation of minimally invasive approaches (including active surveillance) for managing tumor recurrence after primary ablation.

Two authors independently performed article screening, selection, and data extraction. The senior author supervised the process and resolved any disagreements. This approach ensures a transparent and reliable review process, minimizing potential biases and enhancing the credibility of our conclusions.

## 3. Results

As mentioned above, we extracted and analyzed the data of 364 patients from 29 studies. These patients underwent minimally invasive modalities (including active surveillance) for management of tumor recurrence following primary renal mass ablation (Table 1). Six studies focused exclusively on laparoscopic or robotic-assisted partial or radical nephrectomy as a secondary salvage approach; ten studies evaluated repeat ablation as the primary modality; and thirteen studies investigated both minimally invasive nephrectomy (partial or radical) and repeat ablation for recurrence management. In eight studies, active surveillance was chosen as the management option for some patients while other options were selected for the rest of patients.

**Table 1 cancers-17-00974-t001:** Summary of studies (*n* = 29) addressing minimally invasive or active surveillance approach for management of failures after primary ablation of small renal mass. Some studies presented multiple modalities.

1	Study Type	Mean Age (Year)	Tumor Size (cm)	Tumor Histology	Mean Time to Recurrence (Month)	Secondary Treatment Modality (No of Cases)	Outcome	Mean Follow-Up (Month)
Gupta et al., 2009 [58]	Retrospective Cohort	55.5	2	RCC	5.5	RFA (2)	2 NED	20.3
Ferakis et al., 2010 [57]	Prospective Cohort	62.3	4.46	-	32	RFA (3)	NED	58.66
Liu et al., 2017 [61]	Retrospective Cohort	-	>4	ccRCC	-	RFA (5)	4 NED 1 Rec	77
Lorber et al., 2014 [64]	Retrospective Cohort	-	2.7	2 ccRCC 1 RCC	86	RFA (3)	NED	65.6
Matsumoto et al., 2005 [65]	Retrospective Cohort	-	1	RCC	<6	RFA (1)	Rec	18
Ji et al., 2016 [59]	Retrospective Cohort	72	3.2	ccRCC	20	RFA (1)	NED	48
Psutka et al., 2013 [70]	Retrospective Cohort	-	<7	RCC	30	RFA (6)	5 NED 1 Rec	45
						AS (6)		
Stern et al., 2007 [72]	Retrospective Cohort	-	<4	ccRCC	18	RFA (1)	NED	12
Long et al., 2017 [63]	Retrospective Cohort	-	-	RCC	13.1	RFA (13) CA (2)	14 NED 1 Rec	38
Tracy et al., 2010 [74]	Retrospective Cohort	-	-	RCC	9.66	RFA (2) CA (1) AS (1)	2 NED 1 Rec Death	-
Caputo et al., 2017 [52]	Retrospective Cohort	-	-	-	-	CA (2)	1 NED 1 Rec	30.1
						AS (2)		
Emara et al., 2014 [55]	Prospective cohort		-	-	-	CA (2)	NED	31.3
Murray et al., 2019 [66]	Retrospective Cohort	-	<4	5 RCC 1 RO	-	CA (6)	5 NED 1 Rec	53
Okhunov et al., 2016 [67]	Retrospective Cohort	64.8	2.4	14 ccRCC 3 pRCC 4 chRCC	32	CA (20)	17 NED 3 Rec	30
Patel et al., 2020 [69]	Retrospective Cohort	74	<4	-	-	CA (5)	NED	42.7
Yanagisawa et al., 2022 [75]	Retrospective Cohort	74	<7	20 ccRCC 2 pRCC	-	CA (22)	20 NED 2 Rec	33
						AS (3)		
Rembeyo et al., 2020 [71]	Retrospective Cohort	-	-	-	13	CA (6) AS (5)	6 NED 2 Deaths	29
Sundelin et al. 2019 [73]	Retrospective Cohort	66.5	3	55 ccRCC 5 pRCC 7 chRCC 5 not specified	11.44	CA (72)	31 NED 38 Rec 3 Lost in F/U	23
Breen et al., 2013 [51]	prospective Cohort	67	-	RCC	12	Ablation (1)	NED	18
Pandolfo et al., 2023 [68]	Retrospective Cohort	-	-	-	-	Ablation (8)	NED	45
Yang et al., 2013 [76]	Retrospective Cohort	-	3.36	RCC	12.33	Ablation (3) AS (1)	NED	70.6
Attawettayanon et al., 2024 [50]		64.8	2.3	-	28	Ablation (12)	7 NED 5 Rec	53
	Retrospective Cohort					RN (2)	Unknown	
Loloi et al., 2020 [62]	Retrospective Cohort	-	3.1	51 ccRCC 13 pRCC 2 chRCC 3 benign 8 non-diagnostics	13.7	Ablation (50) AS (14)	38 NED 12 Rec	28
		-				PN (5) RN (8)	Unknown	
Long et al., 2017 [63]	Retrospective Cohort	-	-	RCC	13.1	PN (1) RN (2)	NED	38
Patel et al., 2020 [69]	Retrospective Cohort	74	< 4	-	-	PN (1) RN (3)	NED	42.7
Rembeyo et al., 2020 [71]	Retrospective Cohort	-	-	-	13	RPN (1) RRN (2)	NED	29
Duffey et al., 2012 [54]	Retrospective Cohort	-	-	RCC	24.5	RPN (2) LRN (1) AS (1)	NED(1) Unknown (2) Death	70
Martini et al., 2021 [12]	Retrospective Cohort	78	<4	RCC	26	RPN (35)	NED	43
Feng et al., 2024 [56]	Case Series	59	2.5	RCC	-	RPN (1)	NED	24
Ferakis et al., 2010 [57]	Prospective Cohort	58	5.1	pRCC	12	RN (1)	NED	78
Gupta et al., 2009 [58]	Retrospective Cohort	58	2.65	ccRCC	16.5	RN (2)	NED	20.3
Zhou et al., 2021 [77]	Retrospective Cohort	74	3.4	RCC	<24	RN (2)	NED	9.5
Caputo et al., 2017 [52]	Retrospective Cohort	-	-	-	-	LRN (1)	NED	30.1
Lorber et al., 2014 [64]	Retrospective Cohort	-	4	ccRCC	42	LRN (1)	NED	65.6
Yanagisawa et al., 2022 [75]	Retrospective Cohort	74	<7	-	-	RRN (1)	Unknown	33
Chang et al., 2015 [53]	Retrospective Cohort	-	-	RCC	-	LRN (2)	NED	67.6
Stern et al., 2007 [72]	Retrospective Cohort	-	<4	ccRCC	24	LRN (1)	NED	12
Klatte et al., 2011 [60]	Retrospective Cohort	-	2.5	ccRCC	14	LRN (4)	NED	-
Ji et al., 2016 [59]	Retrospective Cohort	58.5	3.5	ccRCC	15	LRN (2)	1 NED 1 Death	78
Tracy et al., 2010 [74]	Retrospective Cohort	-	-	RCC	12	LRN (1)	NED	36

NED: No Evidence of Disease; RFA: Radiofrequency Ablation; CA: Cryoablation; Rec: Recurrence; AS: Active Surveillance; PN: Partial Nephrectomy; RN: Radical Nephrectomy; L: Laparoscopic; R: Robotic; RCC: Renal Cell Carcinoma; ccRCC: Clear-Cell Renal Cell Carcinoma; chRCC: Chromophobe Renal Cell Carcinoma; pRCC: Papillary Renal Cell Carcinoma; RO: Renal Oncocytoma.

All the patients included in this review had tumor recurrence after primary ablative therapy (*n* = 364). As shown in Table 2, most of these patients had undergone CA (198 cases: 54.3%) and RFA (123 cases: 33.7%) as their primary method of ablation.

For the management of tumor recurrence after primary ablation among the 364 patients included in this review, 249 patients (68.4%) underwent a second ablation therapy, 82 cases (22.5%) were treated with laparoscopic/robotic partial or radical nephrectomy, and 33 cases (9%) were managed with active surveillance (Table 3).

Of the 249 patients who underwent re-ablation for tumor recurrence, 179 (71.9%) had no evidence of disease (recurrence) during follow-up. Tumor recurrence was observed in 67 cases (26.9%), while the procedural outcomes could not be determined in 3 (1.2%) patients. Among 82 cases treated with laparoscopic/robotic partial or radical nephrectomy, the outcomes for 18 cases (21.9%) were either unavailable or the patients were lost to follow-up. Among the remaining 64 patients with reported treatment outcome, only one patient experienced recurrence and succumbed to the disease, while 63 patients were successfully treated with no recurrence reported during follow-up period.

Regarding the 33 patients managed with active surveillance, there were significant limitations in obtaining follow-up details and outcomes. Of the reported cases, four patients died during follow-up, four eventually underwent surgery, and 25 remained under active monitoring. Comprehensive information on secondary treatment outcomes is presented in Table 4.

## 4. Discussion

Since 2017, the AUA guidelines have recommended ablation therapy as an alternative option for managing cT1a solid renal masses (<4 cm in size) [78]. These SRMs are generally characterized by slow growth, a low-grade histology, and a minimal tendency for metastasis [79]. While partial nephrectomy remains the standard of care for cT1a renal masses, thermal ablation and active surveillance are particularly viable options for managing the disease in older patients with significant comorbidities [78].

As for the common urologic and nonurological complications related to these modalities described by the AUA meta-analysis, partial nephrectomy was associated with a significantly higher rate of urologic complications (9%) compared to CA (5%) and RFA (6%). Major nonurological complications, on the other hand, were comparable across these modalities (~5%). Additionally, the transfusion rate was significantly higher in LPN (6%) relative to CA (3%) and RFA (2%). Notably, the reintervention rates were statistically similar between partial nephrectomy and ablation modalities (3%) [78].

According to the current AUA guidelines, patients undergoing ablative procedures should have contrast-enhanced imaging both before and within six months after treatment if biopsy results indicate malignant or nondiagnostic tumors [78]. At most institutions, computed tomography (CT) or magnetic resonance imaging (MRI) is performed within the first month after the ablation to detect residual tumors (i.e., failed ablation) and to establish a baseline imaging of the ablation zone. Protocols for detecting recurrence after the negative initial follow-up imaging vary by institution. Typically, three to four imaging studies are performed during the first year, followed by imaging every 6 to 12 months in subsequent years [80]. For monitoring patients after primary ablation, AUA guidelines recommend a low-risk postoperative follow-up protocol that involves medical history reviews, physical examination, laboratory testing, abdominal imaging, and chest imaging at specified intervals (6, 12, 24, 36, 48, 60, 72–84, and 96–120 months) [3].

In our review, about 65% of recurrences were identified between 6 and 30 months after primary ablation. However, several studies documented recurrences that may occur as late as 54 months, highlighting the importance of long-term imaging to detect late recurrence after primary ablation. Stewart et al. demonstrated that adherence to the 2014 AUA guideline imaging recommendations for low-risk cases could miss at least 60% of recurrences if follow-up is limited to three years [81].

Interpreting findings on post-procedure CT and MRI scans requires expertise in recognizing the unique imaging features of the ablation zone. For instance, benign peri-ablation enhancement, commonly observed on post-procedure imaging, may persist for months before resolving. Additionally, homogenous contrast enhancement of the ablation zone immediately after RFA can mimic tumor recurrence [82,83]. One study reported that 1 in 5 patients (20%) treated with CA showed peripheral rim enhancement of the treatment zone at three months after the ablation, which decreased by approximately 5% per year during follow-up. By the end of the study, only 1.8% of their patients remained suspicious of having residual tumors [25].

Therefore, active surveillance can play a crucial role after renal ablation, enabling the monitoring of imaging findings and the distinguishing of benign post-treatment phenomena and true tumor recurrence. To avoid overtreatment, Breda et al. proposed one-year surveillance as a safe management strategy for patients presenting with early enhancement on scans following CA or RFA [84].

There is currently no consensus on the optimal management modality for tumor recurrence following ablative therapies. Re-ablation, partial or radical nephrectomy, and active surveillance remain the primary management strategies for addressing post-ablation tumor recurrence [10]. All cases included in this review involved patients diagnosed with tumor recurrence after primary ablation who opted for active surveillance or a minimally invasive salvage modality such as re-ablation or laparoscopic/robotic partial or radical nephrectomy. Higher comorbidity indices among patients undergoing primary ablation and concerns about adhesions and surgical complications may lead many clinicians to favor ablation for managing recurrent tumors after primary ablation.

In this review, of the 364 analyzed patients, 249 (68.4%) underwent salvage ablation; the majority (>90%) of these patients were followed for more than 20 months (range 10–132 months). Of the 249 patients who underwent re-ablation for tumor recurrence following primary ablation, 179 (71.8%) patients achieved successful second ablation, with no evidence of disease during follow-up. Sixty-seven patients (26.9%) experienced new recurrences following secondary ablation, while three were lost to follow-up.

The 71.8% success rate of second ablation observed in our review closely aligns with the 76% success rate (38 out of 50 cases) reported in a prior cohort study on the management of residual or recurrent disease following the thermal ablation of renal cortical tumors [62]. Interestingly, four studies included in our review documented outcomes of a third retreatment attempt in 12 patients with recurrences after the second ablation. Among these, two patients underwent successful salvage nephrectomy, seven patients achieved successful tertiary ablations, and three cases were observed after a failed tertiary ablation.

While the overall success rate of second ablations is promising, it is imperative to counsel patients on the chance of failure and the need for proceeding with partial/radical nephrectomy. This approach ensures informed decision-making before treatment.

In the case of a nephrectomy after primary ablation, although partial nephrectomy as a nephron-sparing procedure is often prioritized, post-ablation fibrosis, adhesions, and associated complications may necessitate intraoperative conversion to radical nephrectomy. These patients should be counseled about the complexity of partial nephrectomy and the potential need for conversion to radical nephrectomy [37].

Notably, despite these concerns, 82 (22.5%) patients in the current series underwent successful laparoscopic/robotic partial or radical nephrectomy, comprising 46 robotic partial nephrectomy (56.1%) and 36 robotic/laparoscopic radical nephrectomy (43.9%) cases. Most salvage nephrectomy cases (~77%) in this review were followed for at least 30–50 months. Among the 64 cases with reported outcomes, all but one patient (who died of cancer) were successfully treated with no evidence of disease during follow-up. For example, Martini et al., reported on 35 patients with tumor recurrence after primary ablation, all of whom were managed with robotic partial nephrectomy, achieving successful outcomes with no instances of recurrence during their follow-up [12].

Overall, while salvage partial or radical nephrectomy in post-ablation settings might be associated with some technical challenges, it can be performed safely in carefully selected patients and yields excellent postoperative outcomes. Future studies directly comparing the outcomes of re-ablation versus minimally invasive nephrectomy are necessary to provide more robust evidence supporting the optimal salvage treatment modality in this setting.
**Clinical decision-making for tumor recurrence after ablation**

When determining the optimal treatment for tumor recurrence following primary ablation, a thorough assessment of the patient’s overall health, comorbidities, tumor characteristics, and procedural considerations is paramount. Robotic partial nephrectomy, when performed by experienced surgeons, offers a high rate of disease control and remains the preferred minimally invasive nephron-sparing approach. However, in patients with significant comorbidities or those less suitable for surgery, re-ablation presents a viable, less invasive alternative with an acceptable success rate. Radical nephrectomy is generally reserved for cases where tumors exhibit adverse features or have high nephrometry scores, necessitating a more aggressive approach.

Active surveillance plays a pivotal role in evaluating both patient status and tumor dynamics, ensuring timely intervention when clinically warranted. This approach is particularly beneficial for patients with small renal masses and substantial comorbidities, where the risks of immediate intervention may outweigh potential benefits. Additionally, surveillance helps differentiate benign post-treatment imaging findings from true tumor recurrence, thereby optimizing patient management and reducing unnecessary procedures. Given the complexity of decision-making in tumor recurrence, individualized management strategies tailored to patient and tumor-specific factors remain essential.

Our review study has several limitations. The scarcity of studies specifically reporting on patients with local recurrence following primary ablation required extensive efforts to identify and analyze matching cases in the literature. A significant number of studies had to be excluded due to insufficient outcome data or the inability to extract relevant case outcomes from pooled datasets. Furthermore, the available data sometimes lacked granularity; some of the studies only provided the number of cases and their outcomes without critical information, such as tumor size, which limited our ability to perform detailed analyses. Variability in physician expertise and the method of ablation procedures may also have contributed to heterogeneity in the reported outcomes. Due to the rarity of the condition, we included and analyzed various retrospective and prospective cohorts, as well as case series, in this systematic review. However, the inherent heterogeneity of samples and study designs remains a common limitation of many systematic reviews, potentially affecting the generalizability of findings. Despite these limitations, this review collected a large series of patients who underwent various salvage minimally invasive treatment modalities after failed primary ablation.

## 5. Conclusions

This review highlights the outcomes of various minimally invasive management strategies for tumor recurrence following the primary ablation of small renal masses, including repeat ablation, laparoscopic/robotic partial and radical nephrectomy, and active surveillance. Our findings underscore the efficacy of a second thermal ablation as a salvage treatment, achieving disease control in more than 70% of patients, while minimally invasive partial or radical nephrectomy demonstrated excellent oncological control (>98%) in carefully selected patients despite technical challenges. Active surveillance can also be considered as a viable option for tumors with indolent features in patients with comorbidities. Surveillance is also a non-invasive approach to distinguish benign post-treatment changes from true recurrence. The variability in timing of recurrence following primary ablation (e.g., the chance of recurrence beyond 30 months) highlights the critical importance of long-term follow-up after ablation to optimize patient outcomes. Longitudinal studies comparing nephrectomy versus re-ablation with extended follow-up are essential to further define the role of these strategies and establish standardized protocols for managing tumor recurrence after primary ablation.

## Figures and Tables

**Figure 1 cancers-17-00974-f001:**
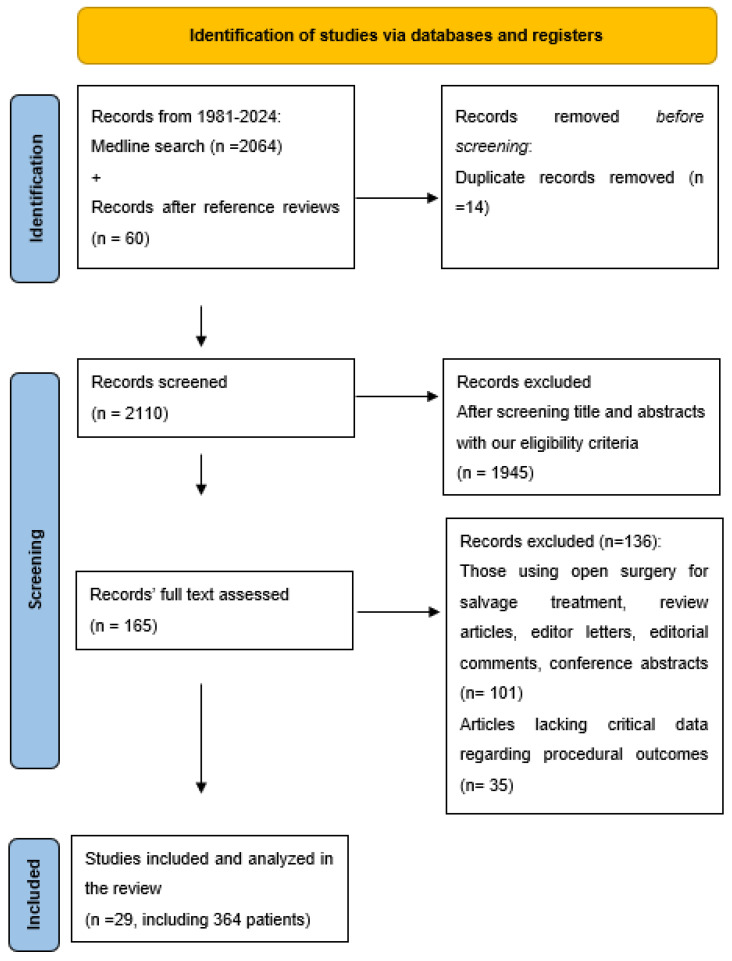
Flow chart of preferred reporting items for systematic reviews and meta-analyses (PRISMA).

**Table 2 cancers-17-00974-t002:** Primary ablative therapy approaches for management of small renal mass in 364 patients who experienced recurrence during follow-up period.

Primary Ablation Approach	Number (%) of Patients
CA	198 (54.3%)
RFA	123 (33.7%)
HIFU	19 (5%)
MVA	2 (0.5%)
Not specified	23 (6%)
Total	364

CA: cryoablation; RFA: radiofrequency ablation; MWA: microwave ablation; HIFU: high-intensity focused ultrasound.

**Table 3 cancers-17-00974-t003:** Secondary treatment modalities for management of failures after primary ablation of small renal mass in 364 patients. CA: cryoablation; RFA: radiofrequency ablation.

Secondary Treatment	Number (%) of Cases	
Re-ablation	249 (68.4%)	RFA: 37 (14.9%) CA: 138 (55.4%) Not Specified: 74 (29.7%)
Laparoscopic/Robotic Partial or Radical Nephrectomy	82 (22.5%)	Partial Nephrectomy: 46 (56.1%) Radical Nephrectomy: 36 (43.9%)
Active Surveillance	33 (9%)	
Total	364	

**Table 4 cancers-17-00974-t004:** Outcomes of various secondary treatment modalities for management of tumor recurrence after primary ablation (*n* = 364).

**Re-Ablation *n* = 249**	
RFA	NED: 33
	Recurrence: 4
CA	NED: 89
	Recurrence: 46
	Unknown: 3
Not specified	NED: 57
	Recurrence: 17
Total	NED: 179 (71.9%)
	Recurrence: 67 (26.9%)
	Unknown: 3
**Laparoscopic/robotic partial or radical nephrectomy *n* = 82**	
Robotic partial nephrectomy (*n* = 46)	NED:40
	Unknown: 6
Robotic/laparoscopic radical nephrectomy (*n* = 36)	NED:23
	Recurrence: 1
	Unknown: 12
Total	NED: 63
	Recurrence: 1
	Unknown: 18
**Active surveillance *n* = 33**	
Surveillance	Surgery: 4
	Death: 4
	Surveillance: 25

NED: No evidence of disease; CA: cryoablation; RFA: radiofrequency ablation.

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
