# Peer review of "Minimally Invasive Salvage Approaches for Management of Recurrence After Primary Renal Mass Ablation"

_cancers, 2025, doi:10.3390/cancers17060974_

Round 1

Reviewer 1 Report

Comments and Suggestions for Authors

This review evaluates salvage treatments options for recurrences follwing ablative treatments for small renal masses. The salvage options including outcomes are clearly demonstrated.

In addition tot he presented data, an overview of the recurrence rates related to the primarily applied ablative treatments would be of interest. Although retrospective, possible differences in recurrence rates after first line treatments with  RFA, MWA, or CA would impact on follow-up strategies and choice of salvage treatment options. Do the authors have data on this?

Author Response

We appreciate our reviewer's comments on the manuscript. We totally agree with our reviewer that there might be a difference in the outcomes of various renal ablation modalities. Based on this comment we added a paragraph to the introduction section: "CA and RFA show comparable recurrence rates, with CA exhibiting local recurrence rates of about 7.2%. RFA recurrence rates range from 4.2% in T1a to 14.3% in T1b tumors, with tumor stage being a key predictor of recurrence risk. Microwave ablation (MWA) demonstrates lower recurrence rates (2–5% at 1 year) compared to CA but similar rates at 5 years, likely due to its higher intertumoral temperature and larger ablation zone [8]."

Reviewer 2 Report

Comments and Suggestions for Authors

This is review for the re treatment alternatives after minimal invasive management of kidney tumor. First of all I must state that this is an important topic that it is interesting for the readers. Nevetheless I have some remarks

1)It is important for the readers to know what kind of review is this. Is this a systematic, a narrative or simple review? Since the Prisma guidelines have been followed then it is most probably systematic but it must be stated and of course reinforced

2)We see studies but we dont see what kind of studies are included and of course how many cases each study has included. In the table that summarizes studies authors must provided data about the type of study (retro, prospective etc). I also think that case series must have been included and this diminishes the strength of the results but nevertheless we must now

3)The main limitations of the study are not the ones that are stated. The main limitations are basically the quality of the included studies, the low number of patients in some of these studies and the lack of statistical analysis of the results.

4)I think that the discussion must be re written in order to give the reader a more easy and critical approach to the results. What and when must be done in patients with relapse of the tumor

Author Response

1)It is important for the readers to know what kind of review is this. Is this a systematic, a narrative or simple review? Since the Prisma guidelines have been followed then it is most probably systematic but it must be stated and of course reinforced.

We appreciate our reviewers’ comments on this manuscript. Since this study is a systematic review, the corresponding line has been added to the manuscript to better reflect it.

2)We see studies but we don't see what kind of studies are included and of course how many cases each study has included. In the table that summarizes studies authors must provided data about the type of study (retro, prospective etc). I also think that case series must have been included and this diminishes the strength of the results but nevertheless we must know

Comment 2- Great suggestion! We revised the by the table 1 based on this comment. The type of included studies as well as the number of cases were shown.  We also added a line to the article to reflect this change:

"Out of the 29 included studies, 24 were retrospective cohort studies, 4 were prospective cohort studies, and one was a case series (page 3 last line)."

3)The main limitations of the study are not the ones that are stated. The main limitations are basically the quality of the included studies, the low number of patients in some of these studies, and the lack of statistical analysis of the results.

Very true! Thank you for this comment. We added address this limitation in the discussion part (Page 11).

"Due to the rarity of the condition, we included and analyzed various retrospective and prospective cohorts, as well as case series, in this systematic review. However, the inherent heterogeneity of samples and study designs remains a common limitation of many systematic reviews, potentially affecting the generalizability of findings."

4)I think that the discussion must be re written in order to give the reader a more easy and critical approach to the results. What and when must be done in patients with relapse of the tumor

Based on this comment we revised the discussion part to enhance readability and emphasize the decision-making parameters.

"Clinical decision-making for tumor recurrence after ablation

When determining the optimal treatment for tumor recurrence following primary ablation, a thorough assessment of the patient’s overall health, comorbidities, tumor characteristics, and procedural considerations is paramount. Robotic partial nephrectomy, when performed by experienced surgeons, offers a high rate of disease control and remains the preferred minimally invasive nephron-sparing approach. However, in patients with significant comorbidities or those less suitable for surgery, re-ablation presents a viable, less invasive alternative with an acceptable success rate. Radical nephrectomy is generally reserved for cases where tumors exhibit adverse features or have high nephrometry scores, necessitating a more aggressive approach.

Active surveillance plays a pivotal role in evaluating both patient status and tumor dynamics, ensuring timely intervention when clinically warranted. This approach is particularly beneficial for patients with small renal masses and substantial comorbidities, where the risks of immediate intervention may outweigh potential benefits. Additionally, surveillance helps differentiate benign post-treatment imaging findings from true tumor recurrence, thereby optimizing patient management and reducing unnecessary procedures. Given the complexity of decision-making in tumor recurrence, individualized management strategies tailored to patient and tumor-specific factors remain essential."

Reviewer 3 Report

Comments and Suggestions for Authors

in this review, the authors analyse the outcomes of various minimally invasive strategies (repeat ablation, laparoscopic or robotic partial/ radical nephrectomy, and active surveillance) in  patients with recurrence of cancer after primary renal ablation.

overall, the paper il clear and provide a comprehensive evaluation of the state of the art in this field. therefore, it can be very useful for readers since , to date, no consensus is available on the optimal minimally invasive salvage approach for managing recurrence after primary ablation

of note, apart from efficacy, it would be useful to have some data about safety (complications type and rates) after the different approaches

Author Response

in this review, the authors analyse the outcomes of various minimally invasive strategies (repeat ablation, laparoscopic or robotic partial/ radical nephrectomy, and active surveillance) in  patients with recurrence of cancer after primary renal ablation. overall, the paper is clear and provide a comprehensive evaluation of the state of the art in this field. therefore, it can be very useful for readers since , to date, no consensus is available on the optimal minimally invasive salvage approach for managing recurrence after primary ablation of note, apart from efficacy, it would be useful to have some data about safety (complications type and rates) after the different approaches. 

Reply

Thank you for reviewing our manuscript. Per this comment we added a paragraph to discuss the potential complication rates of ablation versus partial nephrectomy. (Discussion section- Paragraph 2)

"As for the common urologic and nonurological complications related to these modalities described by the AUA meta-analysis, partial nephrectomy was associated with a significantly higher rate of urologic complications (9%) compared to CA (5%) and RFA (6%). Major nonurological complications, on the other hand, were comparable across these modalities (~5%). Additionally, the transfusion rate was significantly higher in LPN (6%) relative to CA (3%) and RFA (2%). Notably, the reintervention rates were statistically similar between partial nephrectomy and ablation modalities (3%) [78]."

Reviewer 4 Report

Comments and Suggestions for Authors

With the increasing acceptance of renal mass ablation as an option for the management of SRM in a selected group of patients, how to treat tumor recurrence has become a clinical relevant question. In this review, the authors found out that re-ablation is a safe approach with satisfactory outcome compared to other minimal invasive options as well as active surveillance. It is a well written manuscript and of interest to the wide urology community. Some minor comments

(1) Could authors comment on if the same ablation modality was used for re-ablation as for primary ablation? 

(2) Could authors summarize the selection criteria for re-ablation used across studies as oppose to partial/radical nephrectomy?

(3) Table 1 - please clarify if "Time to recurrence" is median time?

Author Response

With the increasing acceptance of renal mass ablation as an option for the management of SRM in a selected group of patients, how to treat tumor recurrence has become a clinical relevant question. In this review, the authors found out that re-ablation is a safe approach with satisfactory outcome compared to other minimal invasive options as well as active surveillance. It is a well written manuscript and of interest to the wide urology community. Some minor comments

(1) Could authors comment on if the same ablation modality was used for re-ablation as for primary ablation?

Thank you very much for reviewing our manuscript. Very thoughtful comment! Unfortunately, in many studies we couldn't find the specific type of "both" primary and salvage ablation. Many of them clearly mentioned either of them. some of them simply used the term "reablation" without giving more details, so we prefer not to make any assumptions and report it as it was. We can assume that, in studies in which modalities were specified, the same modalities were mostly used in re-ablation if the second ablation was done at the same facility. However, the modality could have changed in the retreatment for those referred from other facilities. As noted in the discussion part this is a limitation of current study.  

(2) Could authors summarize the selection criteria for re-ablation used across studies as oppose to partial/radical nephrectomy?

Thank you for this comment.  Out of 29 studies matching our criteria, only 11 studies compared ablation vs nephrectomy, and of these 11 studies, the majority were retrospective studies. Overall, in most cases, the decision between ablation or surgeries was described as “The choice of surgical approach was individualized, based on discussions between the patient and surgeon about the advantages and disadvantages of PN vs ablation” We think that this review can provide some information regarding the prevalence and the outcome of each approach in salvage setting. This information can be helpful for patient counseling.

Per this comment, we revised the discussion part to enhance readability and emphasize the decision-making parameters (Page 11). 

"Clinical decision-making for tumor recurrence after ablation

When determining the optimal treatment for tumor recurrence following primary ablation, a thorough assessment of the patient’s overall health, comorbidities, tumor characteristics, and procedural considerations is paramount. Robotic partial nephrectomy, when performed by experienced surgeons, offers a high rate of disease control and remains the preferred minimally invasive nephron-sparing approach. However, in patients with significant comorbidities or those less suitable for surgery, re-ablation presents a viable, less invasive alternative with an acceptable success rate. Radical nephrectomy is generally reserved for cases where tumors exhibit adverse features or have high nephrometry scores, necessitating a more aggressive approach.

Active surveillance plays a pivotal role in evaluating both patient status and tumor dynamics, ensuring timely intervention when clinically warranted. This approach is particularly beneficial for patients with small renal masses and substantial comorbidities, where the risks of immediate intervention may outweigh potential benefits. Additionally, surveillance helps differentiate benign post-treatment imaging findings from true tumor recurrence, thereby optimizing patient management and reducing unnecessary procedures. Given the complexity of decision-making in tumor recurrence, individualized management strategies tailored to patient and tumor-specific factors remain essential."

(3) Table 1 - please clarify if "Time to recurrence" is median time?

When reporting time to recurrence in Table 1, mean values were used to report. If the mean values were not provided or could not be calculated, time was reported as < --months.  Per this comment, Mean value was added to the table for better clarification.

Round 2

Reviewer 2 Report

Comments and Suggestions for Authors

The paper is ready for publication